# Classification of the ICU Admission for COVID-19 Patients with Transfer Learning Models Using Chest X-Ray Images

**DOI:** 10.3390/diagnostics15070845

**Published:** 2025-03-26

**Authors:** Yun-Chi Lin, Yu-Hua Dean Fang

**Affiliations:** 1Department of Biomedical Engineering, University of Alabama at Birmingham, Birmingham, AL 35294, USA; ycl2@uab.edu; 2Department of Radiology, Heersink School of Medicine, University of Alabama at Birmingham, Birmingham, AL 35233, USA; 3Department of Neurology, Heersink School of Medicine, University of Alabama at Birmingham, Birmingham, AL 35233, USA

**Keywords:** COVID-19, ICU admission, chest X-ray, transfer learning, deep learning, dataset expansion, Grad-CAM, medical image analysis, DenseNet121

## Abstract

**Objectives**: Predicting intensive care unit (ICU) admissions during pandemic outbreaks such as COVID-19 can assist clinicians in early intervention and the better allocation of medical resources. Artificial intelligence (AI) tools are promising for this task, but their development can be hindered by the limited availability of training data. This study aims to explore model development strategies in data-limited scenarios, specifically in detecting the need for ICU admission using chest X-rays of COVID-19 patients by leveraging transfer learning and data extension to improve model performance. **Methods**: We explored convolutional neural networks (CNNs) pre-trained on either natural images or chest X-rays, fine-tuning them on a relatively limited dataset (COVID-19-NY-SBU, *n =* 899) of lung-segmented X-ray images for ICU admission classification. To further address data scarcity, we introduced a dataset extension strategy that integrates an additional dataset (MIDRC-RICORD-1c, *n* = 417) with different but clinically relevant labels. **Results**: The TorchX-SBU-RSNA and ELIXR-SBU-RSNA models, leveraging X-ray-pre-trained models with our training data extension approach, enhanced ICU admission classification performance from a baseline AUC of 0.66 (56% sensitivity and 68% specificity) to AUCs of 0.77–0.78 (58–62% sensitivity and 78–80% specificity). The gradient-weighted class activation mapping (Grad-CAM) analysis demonstrated that the TorchX-SBU-RSNA model focused more precisely on the relevant lung regions and reduced the distractions from non-relevant areas compared to the natural image-pre-trained model without data expansion. **Conclusions**: This study demonstrates the benefits of medical image-specific pre-training and strategic dataset expansion in enhancing the model performance of imaging AI models. Moreover, this approach demonstrates the potential of using diverse but limited data sources to alleviate the limitations of model development for medical imaging AI. The developed AI models and training strategies may facilitate more effective and efficient patient management and resource allocation in future outbreaks of infectious respiratory diseases.

## 1. Introduction

The Coronavirus disease 2019 (COVID-19) caused by the SARS-CoV-2 virus was declared as a public health emergency of international concern by the World Health Organization (WHO) in January 2020 and has led to over 774 million confirmed cases and 7 million deaths globally as of 2024 [1]. Given the high mutation rates of ribonucleic acid viruses, new variants that are more transmissible or treatment-resistant remain a concern for public health [2]. To date, although COVID-19 is perceived as an endemic illness with seasonal peaks like influenza, its prevalence continues to require ongoing surveillance. For example, simultaneous outbreaks of multiple infectious respiratory diseases including COVID-19, influenza, and respiratory syncytial virus (RSV) can significantly challenge the healthcare system. The combined impact of these respiratory illnesses led to a significant increase of 23% in new hospitalization and 51% in intensive care unit (ICU) admissions globally during November and December in 2023 compared to the previous month [3]. This highlights the need for better patient risk stratification and resource management tools.

Currently, the food and drug administration (FDA)-approved COVID-19 therapeutics include anti-inflammatory agents, antiviral agents, and neutralizing antibody therapies. Treatment choices are based on the severity of the infection and specific risk factors. Early stages of infection, characterized by high viral replication, benefit significantly from antiviral and antibody-based interventions, while the later hyperinflammatory phase necessitates anti-inflammatory treatments such as corticosteroids and immunomodulating therapies [4]. Studies have shown the clear advantages of early interventions for high-risk COVID-19 patients [5,6] through randomized controlled trials; early treatments with recombinant neutralizing monoclonal antibodies significantly reduced the risks of hospitalization or death, particularly for seronegative individuals [7,8,9]. Additionally, a study showed that early Remdesivir treatment in high-risk non-hospitalized patients led to an 87% reduction in hospitalization or death risk [10]. Thus, early and accurate risk stratification is essential in leveraging these therapeutic advancements and minimizing the burden on healthcare systems. Patient outcome may also be significantly improved with accurate patient risk assessments.

Medical imaging, particularly chest X-rays and computed tomography (CT), has played a crucial role in COVID-19 diagnosis and severity assessment. While CT scans provide excellent diagnostic and prognostic performances [11,12,13,14,15,16,17,18], chest X-rays are more commonly and frequently taken of COVID patients because of their accessibility, efficiency, lower radiation dose, and lower costs. Deep learning enables the development of direct and automatic artificial intelligence (AI) models of image processing and interpretation. However, medical imaging AI can be limited due to data availability, as large, well-annotated public datasets are less common due to privacy concerns, curation challenges, and the need for expert annotation [19]. As a result, models trained on small datasets often struggle with generalizability. Transfer learning has been demonstrated as a useful technique for improving performance in data-limited scenarios [20]. Leveraging the knowledge from pre-trained models trained on large datasets reduces the need for extensive labeled data and accelerates convergence [21].

In the fight against COVID-19, chest X-ray-based deep learning algorithms have been explored for diagnosis and severity assessment to support patient management [22,23,24,25]. However, their potential in the early identification of patients that later progress to severe infection, particularly those that require ICU admission, remains relatively underexplored. In an outbreak of infectious diseases that leads to an excessive demand for intensive care for patients with severe symptoms, the early prediction of an individual patient’s outcome can help medical caregivers to allocate medical resources efficiently and prepare for potential public health challenges. Such prognostic models can further benefit patient management by enabling timely therapeutic interventions, potentially preventing deterioration and reducing the need for ICU admissions. Emerging research has highlighted the potential of radiologist-evaluated or AI-derived X-ray severity scores as strong predictors of adverse outcomes in COVID-19 patients [26,27,28,29,30,31]. Although deep learning allows for automatic airspace scoring, training these models still requires large amounts of expert-annotated data, which is often infeasible in data-limited scenarios, especially during the early surge of an infectious disease. A more adaptable approach is an end-to-end model that directly predicts patient outcomes from chest X-rays, bypassing the scoring process. Among the few existing studies focusing on this task, Shamout et al. developed the COVID-GMIC model, using 6449 chest X-rays to predict patient deterioration, including ICU admission, intubation, or in-hospital mortality within 96 h, achieving an area under the receiver operating characteristic (ROC) curve (AUC) of 0.74, which further improved to 0.79 when incorporating clinical laboratory data [32]. Similarly, Li et al. presented a DenseNet121-based model trained on 8357 chest X-rays, achieving AUCs of 0.78, 0.77, 0.76, and 0.76 in predicting ICU admission or intubation at 24, 48, 72, and 96 h, respectively [33]. While these studies demonstrate the potential of AI in patient risk stratification, they were both trained with private and large (*n* > 6000) datasets. This poses several challenges: (1) limited accessibility, as private datasets restrict replication and external validation; and (2) difficulty in implementation during early outbreak scenarios, where large-scale labeled datasets are usually unavailable. This data dependence fundamentally limits the use of these models during the early phase of infectious disease outbreaks.

To address this gap, our study explores techniques for developing AI models in data-limited scenarios using a small, open-access chest X-ray dataset (*n =* 899). We propose an AI-based ICU admission classification model for COVID-19 patients using transfer learning techniques. Specifically, we aim to achieve the following:Compare CNN models pre-trained on natural images vs. X-rays to assess the impact of domain-specific pre-training;Introduce a dataset extension strategy, incorporating an external dataset (*n =* 417) with different yet clinically relevant labels;Evaluate model performance across different training strategies;Use Grad-CAM analysis to examine whether the models focus on lung regions that are physiologically meaningful for severe lung infections.

Through these approaches, we aim to develop a more adaptable and accessible AI model that enhances ICU admission classification in data-limited scenarios to improve real-world applicability in healthcare settings.

## 2. Materials and Methods

### 2.1. Datasets

#### 2.1.1. Overview of the Datasets

In this study, we used two openly available datasets: COVID-19-NY-SBU [34] and MIDRC-RICORD-1c [35]. The primary dataset, COVID-19-NY-SBU, contains chest X-ray images with ICU admission labels as the main outcome of interest for this study. On the other hand, MIDRC-RICORD-1c is employed to expand the training data by providing additional chest X-ray images annotated with airspace disease grading, a highly correlated proxy for severe pneumonia that may lead to intensive care. Combining these datasets serves a dual purpose; it addresses the challenge of data scarcity and leverages the indirectly labeled data in MIDRC-RICORD-1c to evaluate whether they may enhance the model’s ability to learn robust features indicative of severe cases that require ICU admission. Figure 1 illustrates the image selection and exclusion details for both datasets.

#### 2.1.2. COVID-19-NY-SBU

This dataset was acquired by Stony Brook Medicine from inpatients who tested positive on a reverse transcription polymerase chain reaction (RT-PCR) for COVID-19 [34]. Data from a total of 1384 patients were involved in this collection.

In our study, the chest X-ray images served as the input to deep learning models to classify whether ICU admission was required based on ICU room charges and in-hospital mortality. The records were sorted into two classes: ICU, for patients who received or needed intensive care, and non-ICU, for those who did not. We also treated in-hospital mortality as part of the ICU group as we observed that some patients (*n =* 46) who passed away during hospitalization were never transferred to the ICU, potentially due to rapid disease progression, severe complications, or ICU beds being unavailable. By including these patients, we capture a broader spectrum of severe COVID-19 cases.

The COVID-19-NY-SBU dataset did not specify the dates of ICU admission or X-ray acquisition. Therefore, we cannot know if X-rays were taken before or after ICU admission for a given patient. However, under typical workflows, X-ray examinations are usually performed as a first-line diagnostic tool, as an early-stage triage for potential COVID-19 patients. As a result, we only used the first X-ray image of each patient, and required this image to be taken within a week of the patient’s initial hospital visit. For image quality screening, we excluded scans captured in a lateral view instead of the AP (anterior-to-posterior) or PA (posterior-to-anterior) view, as well as cropped or incomplete scans that lacked the full lung fields. After excluding based on the timeframe and the image quality criteria, 899 X-ray images from this dataset (269 ICU, 630 non-ICU) were used for model training and testing.

#### 2.1.3. MIDRC-RICORD-1c

The MIDRC-RICORD-1c dataset was collected and made available by the Radiological Society of North America (RSNA) as a collection of the RSNA International COVID-19 Open Radiology Database (RICORD) in its Medical Imaging Data Resource Center (MIDRC) [35]. The MIDRC-RICORD-1c dataset contains 998 X-ray images from 361 COVID-19 patients across four international sites, annotated with airspace disease grading. The scoring method divides each lung into three zones (six zones total) and grades severity as follows: mild (opacities in 1–2 zones), moderate (opacities in 3–4 zones), and severe (opacities in more than 4 zones). To adapt this grading to our binary ICU admission label, we referenced the literature on airspace severity scoring systems that assess lung involvement extent, which might better align with the RSNA’s zonal criteria. The evidence indicates that patients with extensive lung involvement—such as those with severity scores in the highest quartile or opacities affecting more than 75% of the lung area—face a significantly higher risk of ICU admission or mortality compared to those with milder disease [26,28,30]. Accordingly, we assigned the ICU label exclusively to severe pneumonia cases (defined as opacities in more than 4 lung zones), while categorizing negative, mild, and moderate cases (0–4 zones) as non-ICU. This classification reflects the clinical distinction where severe cases are markedly more critical. After excluding cases with missing grading labels or those not captured in the AP or PA view, a total of 417 chest X-ray images (118 ICU and 299 non-ICU) were included in this study.

### 2.2. Image Preprocessing

#### 2.2.1. Lung Segmentation

The chest X-ray images were first pre-processed to extract the region of the lung. We used a U-net-based segmentation model developed by Illia Ovcharenko on GitHub https://github.com/IlliaOvcharenko/lung-segmentation (accessed on 9 August 2022) to segment the lungs. This model was developed with a pre-trained vgg11 encoder with additional batch normalization layers added to a basic U-net. Subsequently, a bounding box was cropped and placed on each image to locate the lung, as illustrated in Figure 2.

#### 2.2.2. Oversampling and Augmentation

Both the COVID-19-NY-SBU and MIDRC-RICORD-1c datasets were imbalanced. The ratio of ICU to non-ICU subjects was about 1:2.3 in COVID-19-NY-SBU and 1:2.5 in MIDRC-RICORD-1c. To address the data imbalance, we performed an oversampling procedure on the training set by augmenting the data through randomly selecting samples from the minority class and then applying a randomized rotation operation of an angle between 15 and −15 degrees. By oversampling, we extended the number of images of the minority class (ICU) to the same as the majority class (non-ICU). It is important to note that the testing data remained untouched by this oversampling procedure to ensure the validity of our model evaluation.

#### 2.2.3. Histogram Equalization and Normalization

We applied contrast-limited adaptive histogram equalization (CLAHE) to enhance the local contrast and to improve the visibility of anatomical structures and abnormalities on chest X-ray images [36]. For image normalization, the X-ray images were adjusted within intensity ranges suitable for the chosen pre-trained models: [0, 255] for ImageNet and [−1024, 1024] for TorchXRayVision. Subsequently, all images were resized to a uniform resolution of 224 × 224 pixels.

### 2.3. Transfer Learning and Model Architectures

We utilized transfer learning techniques to improve the training of deep networks with relatively small datasets. The model was composed of two major parts: imaging feature extraction and classification. For imaging feature extraction layers, we used two conventional pre-trained CNNs: ImageNet pre-trained with natural images and TorchXRayVision pre-trained with chest X-ray images [37,38]. For both the ImageNet and TorchXRayVision models, we selected DenseNet121 as the model architecture based on its strong performance in COVID-19 detection and severity assessment [32,33,39]. We froze all layers from the bottom except for the last convolutional layer, making it trainable for imaging feature extraction. Beyond conventional CNNs, we also used ELIXR, an image encoder pre-trained on large chest X-ray datasets, to generate embeddings that capture dense, clinically relevant features [40,41]. We extracted the embedding representations (128 × 64) from the ELIXR output. After feature extraction, both DenseNet121 and ELIXR were connected to a classification pipeline consisting of two fully connected layers with dropout and batch normalization layers, as illustrated in Figure 3.

### 2.4. Evaluation

We used a five-fold cross-validation to evaluate the performance of the proposed model with six performance indices—precision, recall, specificity, balanced accuracy, AUC, and the area under the precision–recall curve (PR AUC). To handle the bias from imbalanced testing data, we chose balanced accuracy as the primary measure of accuracy, rather than the conventional overall accuracy. This allowed us to better evaluate model performance for both the majority and minority classes. These parameters were formulated using the confusion matrices as follows:(1)$$Precision=\frac{TP}{TP+FP}$$(2)$$Recall=\frac{TP}{TP+FN}=Sensitivity$$(3)$$Specificity=\frac{TN}{TN+FP}$$(4)$$BalancedAccuracy=\frac{Sensitivity+Specificity}{2}$$

We also utilized gradient-weighted class activation mapping (Grad-CAM) to visualize the regions of importance generated by the DenseNet models after evaluating an image. The concept of Grad-CAM was presented by R. R. Selvaraju et al., and their technique is marked by generating the visual explanation of CNN model evaluation without requiring architectural changes or re-training [42]. The gradient information on the last convolutional layers is obtained to assign a spatial map of importance for the output decisions.

### 2.5. Experiment

Our study evaluated four distinct models (ImageN-SBU, TorchX-SBU, TorchX-SBU-RSNA, and ELIXR-SBU-RSNA), each associated with distinct pre-trained and training datasets:ImageN-SBU: ImageNet pre-trained model re-trained with COVID-19-NY-SBU;TorchX-SBU: TorchXRayVision pre-trained model re-trained with COVID-19-NY-SBU;TorchX-SBU-RSNA: TorchXRayVision pre-trained model re-trained using an extended training dataset containing both the COVID-19-NY-SBU and the MIDRC-RICORD-1c data;ELIXR-SBU-RSNA: X-ray-pre-trained image encoder, with classification layers trained on both the COVID-19-NY-SBU and the MIDRC-RICORD-1c data.

We evaluated the model performances through five-fold cross-validation using the testing data randomly selected from the COVID-19-NY-SBU dataset. For each evaluation fold, we split the data from COVID-19-NY-SBU into training and testing subsets with a ratio of 4:1 before data augmentation. For the TorchX-SBU-RSNA and ELIXR-SBU-RSNA models, we augmented the training subset by incorporating all images from the MIDRC-RICORD-1c dataset as part of the training data. Accounting for data skewness, we then augmented the training data as previously described. No augmentation was applied to the testing data so that all testing data remained unseen to the model to preserve data leakage. Table 1 and Table 2 show the exact distribution of images for each subset for these two data splits.

Our model complexity arises from the deep architecture of the CNN and training strategies designed to optimize performance while addressing class imbalance and overfitting. Using transfer learning, we significantly reduced the total parameters by over 85%, resulting in 1 million trainable parameters in DenseNet121. Given the high computational demand, we implemented the model using the PyTorch v2.1.2 framework on NVIDIA Tesla A100 GPUs for efficient processing. To fine-tune performance, we experimented with optimizers (Adam, SGD), learning rates (0.00001–0.01), batch sizes (16–64), and dropout rates (0.3–0.7). Overfitting was mitigated through regularization with weight decay (1 × 10^−5^–1 × 10^−3^), ReduceLROnPlateau for adaptive learning rate adjustments, and early stopping (patience = 50). During training, the best model weights were saved and reloaded for evaluation.

## 3. Results

### 3.1. Comparison of Performance Between Different Pre-Trained Models

We evaluated the impact of different pre-trained models on the performance of ICU admission classification, with their main difference being the type of training data during the pre-training phase. The ImageN-SBU (pre-trained with natural images) and TorchX-SBU (pre-trained with X-ray images) models were fine-tuned on segmented images from COVID-19-NY-SBU with various combinations of hyperparameters. Table 3 lists the detailed performance comparison between the two models. The ImageN-SBU model had a balanced accuracy of 64% and an AUC of 0.66, while the TorchX-SBU model reached a higher balanced accuracy of 68% and an improved AUC of 0.71. This suggests that pre-training with X-ray-specific data can modestly enhance model performance, as evidenced by an AUC improvement of approximately 0.05. Moreover, precision, recall, specificity, F1 scores, and balanced accuracy were all moderately improved, indicating that the X-ray-pre-trained model can enhance the model’s classification ability. Figure 4 presents the average accuracy and loss from five-fold validation over training epochs, revealing potential overfitting in the ImageN-SBU model while the training of the TorchX-SBU model seems more satisfactory.

### 3.2. Impact of Training Data Expansion on Model Performance

We explored the impact of extending the training dataset by incorporating an additional dataset annotated with airspace disease severity levels in the TorchX-SBU-RSNA model. We also evaluated the ELIXR-SBU-RSNA model, which leverages a pre-trained X-ray embedding encoder. The model performances are summarized in Table 3.

The TorchX-SBU-RSNA model demonstrated a further improved balanced accuracy of 69% and an AUC of 0.77, compared to the TorchX-SBU model’s balanced accuracy of 65% and AUC of 0.71. Specifically, TorchX-SBU achieved a specificity of 71% and a recall of 59%, whereas TorchX-SBU-RSNA further enhanced specificity to 80%, while maintaining a recall of 58%. This suggests that the TorchX-SBU-RSNA model was better at identifying non-ICU cases, while preserving its ability to detect ICU cases. The enhanced specificity of the TorchX-SBU-RSNA model can be attributed to a broader variety of cases included in the extended dataset, which likely allowed the model to learn more relevant features indicative of severe lung infections and to identify critical patterns associated with the need for ICU admission. By accessing a more diverse dataset that captures a wider spectrum of disease severity and relevant imaging features, the model was able to improve its recognition of non-severe cases.

Beyond the conventional CNN models, the ELIXR-SBU-RSNA model demonstrated a very similar performance to the TorchX-SBU-RSNA model, achieving a balanced accuracy of 70% and an AUC of 0.78. Compared to TorchX-SBU-RSNA, it exhibited a slightly higher recall (62% vs. 58%) but a lower specificity (78% vs. 80%).

### 3.3. Grad-CAM Evaluation

The results of the Grad-CAM for evaluating the focal areas significant for ICU admission classification in the model were visually assessed and demonstrated through the representative subjects shown in Figure 5. In general, we observed that the X-ray-pre-trained model put more emphasis on the lungs and reduced the distraction by adjacent regions like the neck, diaphragm, or shoulders. Notably, the TorchX-SBU-RSNA model concentrated more on its heat maps on the lung and relevant areas than the TorchX-SBU model, showing a better ability to focus on the lung pathologies and avoid irrelevant features after training. Since the ELIXR-SBU-RSNA model performs classification based on ELIXR-generated embeddings rather than directly on X-ray images, Grad-CAM results are not available for this model.

## 4. Discussion

In this study, we developed deep learning models using chest X-ray images to classify ICU admissions of COVID-19 patients and evaluated the model performance under different training strategies. In medical imaging AI, the limited availability of diverse training data often poses a significant barrier for model development and validation. In AI model development for chest X-ray-based ICU classification, the current models require large sample sizes and have been trained with private datasets. We therefore seek to develop AI models and strategies that may reduce the demand for large datasets and provide satisfactory model training with publicly available and smaller datasets. Previous studies have shown that transfer learning is a powerful technique for improving the performance of CNN models in medical imaging applications, particularly when training data are limited [24,25,43,44]. In this study, we leveraged natural image and X-ray-pre-trained models for imaging feature extraction, and fine-tuned the models with another smaller X-ray dataset to classify the needs of ICU care. By leveraging transfer learning and unique data expansion techniques, we demonstrated that it is feasible to efficiently develop AI classification models with limited access to training data. This may be of particular interest and usefulness during the initial stages of potential outbreaks in the future, when efficient triage is critical for resource management but available data are still scarce.

In our results, we observe significant overfitting in the ImageNet-pre-trained model compared to the X-ray-pre-trained model, even though we have taken optimization measures such as adjusting or optimizing dropout, learning rate decay, batch normalization, and early stopping to mitigate overfitting. This is likely due to intrinsic data differences, including domain mismatch and limited medical data diversity. Prior studies have shown the challenges of adapting large-scale natural image pre-trained models for domain-specific medical applications [45,46]. Specifically, ImageNet is a large dataset consisting of natural images, and the features learned by models pre-trained on ImageNet are generic and aimed at distinguishing between very diverse categories of visual objects. When fine-tuned for specialized tasks like X-ray analysis, such models may be prone to retaining high-variance feature sensitivity, which could possibly lead to overfitting from capturing noise rather than meaningful patterns. Additionally, X-ray datasets, particularly those used for specific conditions like COVID-19, are often smaller and less diverse compared to ImageNet. This disparity can exaggerate overfitting when a complex model trained with natural image datasets like ImageNet is applied to a smaller, more homogeneous dataset. In our study, the other models pre-trained with X-rays clearly outperformed the ImageN-SBU model, demonstrating that the X-ray-pre-trained models should be adopted as the primary pre-trained models for developing X-ray AI models, especially under limited data availability.

In addition to the finding that X-ray-pre-trained models consistently yielded better and more stable performances, we also demonstrated through Grad-CAM results that ImageNet-based models, as they were pre-trained with natural images, are less efficient in capturing relevant details than medical image-pre-trained models. Additionally, we observed a tendency for the TorchX-SBU-RSNA model to focus on the lower part of lungs in the Grad-CAM heat maps of COVID-19 patients. This pattern is consistent with characteristic chest radiograph findings reported in previous studies on COVID-19 patients [47,48,49,50]. Moreover, in another retrospective study, the appearance of bilateral opacity/consolidation in a peripheral and mid to lower lung zone distribution on CT scans achieved a high specificity of 96.6% in identifying severe COVID-19 infection [51]. The fact that our model’s focus in the Grad-CAM results aligns with these recognized patterns of COVID-19 infections, as shown in the literature, demonstrates that it has improved training performance and abilities to learn relevant features for COVID severity and patient risks.

We further expanded our original training dataset by incorporating another dataset labeled with airspace disease grading to extend the training data. This led to our best-performing models, TorchX-SBU-RSNA and ELIXR-SBU-RSNA, which achieved balanced accuracies of 69–70% and AUCs of 0.77–0.78. Specifically, the extension of the training data increased the specificity from 71% to 80% compared with the TorchX-SBU without data extension. This finding illustrates the benefits of expanding training sets with different yet highly related data labels as a novel approach for chest X-ray imaging AI proposed in this study. Overall, the findings of this study on the advantages of X-ray-pre-trained models and data extension strategies may better guide future studies in scenarios of limited data availability, or when fast training must be obtained at an early stage of investigation. Such strategies may also help to improve model training when larger datasets are available.

Compared to prior imaging AI studies that predicted ICU admissions of COVID-19 patients using chest X-rays, our model achieved competitive results without requiring extensive expert annotations or large private datasets. Some prior studies have utilized established radiologist-evaluated X-ray severity scoring systems alongside deep learning to automate assessments, achieving notable AUCs in predicting adverse events, including ICU admissions, intubations, and deaths [30,31]. Specifically, Chamberlin et al. developed a model that included both COVID and non-COVID patients in emergency departments, achieving an AUC of 0.87 [30]. With the broader inclusion criteria for this study’s cohort, Chamberlin et al.’s model may not be directly comparable to studies that exclusively focus on COVID-19 patients.

On the other hand, some studies have explored direct models that do not require lung disease severity scoring, using chest X-ray and clinical information to predict ICU needs. A model comparison is summarized in Table 4.

One of the leading models was developed by Shamout et al., who utilized 6449 X-ray images to achieve an AUC of 0.74 (PR AUC of 0.44) for predicting ICU admission, intubation, or in-hospital mortality within 96 h, which was improved to 0.79 AUC (0.52 PR AUC) with additional clinical data, such as vital signs and lab tests [32]. Our study, while working solely with much smaller X-ray datasets (COVID-19-NY-SBU, *n =* 899; MIDRC-RICORD-1c, *n =* 417; combined dataset, *n =* 1316) and without requiring lab or clinical data, produced a competitive and comparative AUC of 0.78 and a notably higher PR AUC of 0.63 compared to Shamout et al.’s 0.52. Moreover, our model achieves an improved precision of 56% compared to 24% in Shamout et al.’s model, indicating a substantial reduction in false positives—a critical factor in minimizing unnecessary ICU admissions. Although a 56% precision rate may not yet be sufficient for clinical use, this improvement demonstrates that our proposed training strategy can enhance model performance even in data-constrained scenarios. Another leading model was developed by Li et al. yielded an AUC of 0.76 for predicting ICU admission or intubation within 96 h based on 8357 X-rays [33]. PR AUC data from Li et al.’s model were unavailable for comparison, so we can only comment that our model, trained on a smaller dataset, seems to provide a similar performance to theirs. Overall, the proposed training strategies and model designs in this study have led to similar performances to the currently leading models, while significantly reducing the demand for large datasets.

There are limitations to this study. First, the early detection of ICU admission for high-risk COVID-19 patients is indeed challenging due to the heterogeneous pathologies of progression that may involve various factors beyond acute viral infection and immune responses in the respiratory system. Prior research indicates that age, smoking status, body mass index (BMI), and chronic diseases are all significant risk factors for severe COVID-19, with diabetes being a particularly critical comorbidity [52,53,54,55,56,57]. Building upon these findings, some predictive scoring systems have been developed [58,59,60]. Across a total of 17 studies, 7 factors including age, D-dimer, C-reactive protein, sequential organ failure assessment (SOFA) score, body temperature, albumin, and diabetes have the highest consistency as predictors of COVID-19 severity [52]. Additionally, multiple studies have shown the potential of utilizing laboratory and clinical features to predict prognosis outcomes, such as mortality and ICU admission [61,62,63,64,65,66]. It is reasonable to expect a better predictive performance when such clinical or laboratory information is added to the imaging AI model. Our study is limited in that the publicly shared datasets of COVID-19-NY-SBU and MIDRC-RICORD-1c did not fully provide every subject’s laboratory measurements or clinical records. For instance, in the COVID-19-NY-SBU dataset, more than half of the patients lack complete data for age, BMI, diabetes, D-dimer, and C-reactive protein (639 out of 1384 patients). In future, researchers may consider exploring the integration of more comprehensive clinical data as part of the AI model to enhance prediction accuracy as more data becomes available [67].

Another limitation of our study lies in the imbalanced data distribution. Typically for retrospective imaging AI studies, it is harder to collect positive cases, especially for labels of severe, life-threatening conditions. This imbalance often leads to the lower sensitivity and precision of the model in identifying the positive class [68]. A commonly used approach is to over-sample the minority class for data augmentation by applying spatial transformations such as flipping, rotation, and shifting [69,70]. In this study, the issue of lower sensitivity and precision persisted even though we augmented the minority class. This may be due to the inherent complexity and variability in the features of the images, meaning that the imaging features have not been sufficiently captured and learned through simple transformations alone. Future work may investigate more advanced data augmentation techniques, such as filtering techniques with Gaussian blurs, sharpening, edge detection filters that highlight or suppress features in an image, or noise injection that introduces robustness to variations [71]. Generative Adversarial Network (GAN) augmentation, which can generate more complex and varied training examples [71,72], may also be promising in the more realistic generation of augmented data. In addition, our study’s generalizability is currently limited, as we were only able to access the dataset from a single institution for validation. Validating our models through prospective studies and external datasets would be a valuable next step. With further refinements, our models could potentially serve as useful tools in clinical decision-making for critical care management and contribute to improved patient outcomes and more efficient use of limited medical resources.

## 5. Conclusions

Our study demonstrates that AI models utilizing COVID-19 X-ray images can serve as a potentially valuable tools for the risk assessment and classification of ICU admissions, even with limited data availability. By leveraging X-ray-pre-trained models and incorporating expanded training datasets with clinically relevant labels, we achieved performances comparable to models trained on large private datasets. This highlights the feasibility of rapid AI model development during the early stages of disease outbreaks, where data scarcity poses a critical challenge. Additionally, our findings demonstrate the effectiveness of expanding training data with different but clinically relevant labels. Future work could explore a more diverse range of surrogate indicators for ICU admission, such as oxygen supplementation requirements and ventilator usage, to maximize the utility of small datasets through data integration and enhance model generalizability and performance. Extending this approach to other emerging infectious diseases would also be valuable in assessing its adaptability to new outbreak scenarios. Given the limitations of data availability in early outbreaks, such an approach would facilitate rapid deployment and may help improve medical decision-making, optimize patient outcomes, and efficiently allocate resources in future public health threats.

## Figures and Tables

**Figure 1 diagnostics-15-00845-f001:**
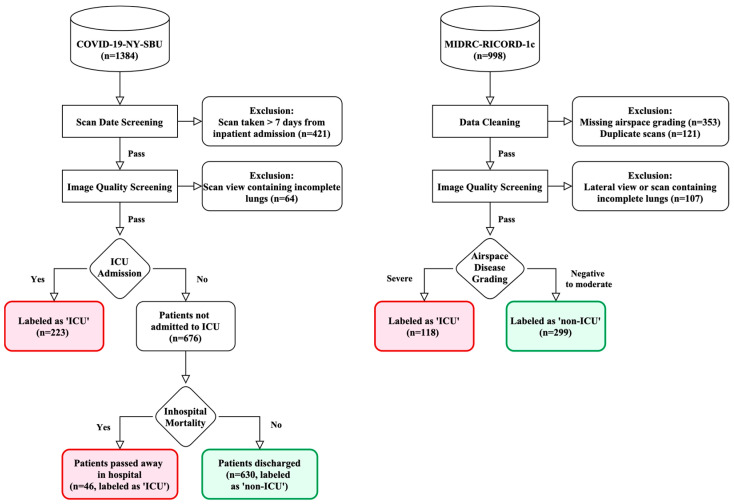
Data exclusion flowchart, depicting the selection process for the COVID-19-NY-SBU and MIDRC-RICORD-1c datasets.

**Figure 2 diagnostics-15-00845-f002:**
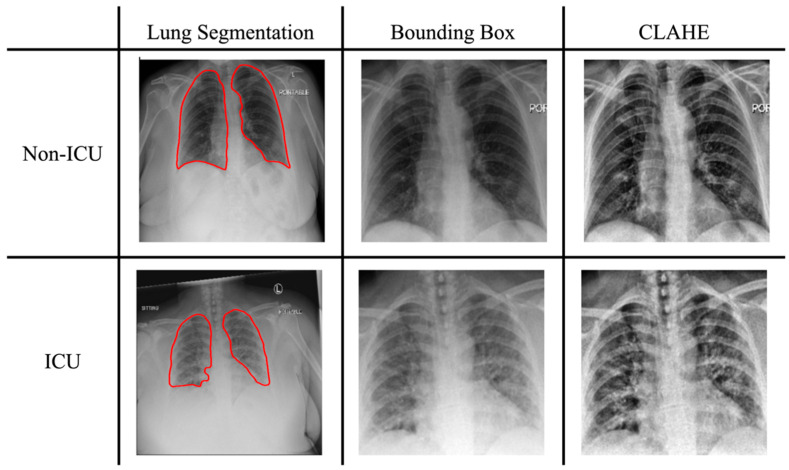
Imaging preprocessing steps applied to sample chest X-rays, including lung segmentation (red outline), bounding box detection, and contrast enhancement using CLAHE.

**Figure 3 diagnostics-15-00845-f003:**
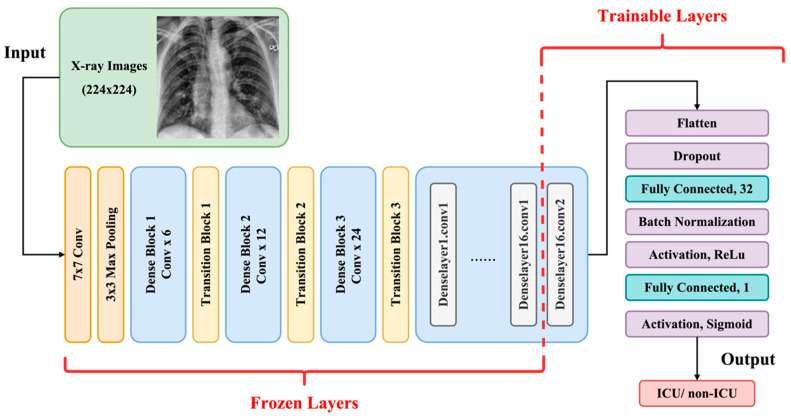
An illustration of the model architecture of transfer learning with pre-trained DenseNet121.

**Figure 4 diagnostics-15-00845-f004:**
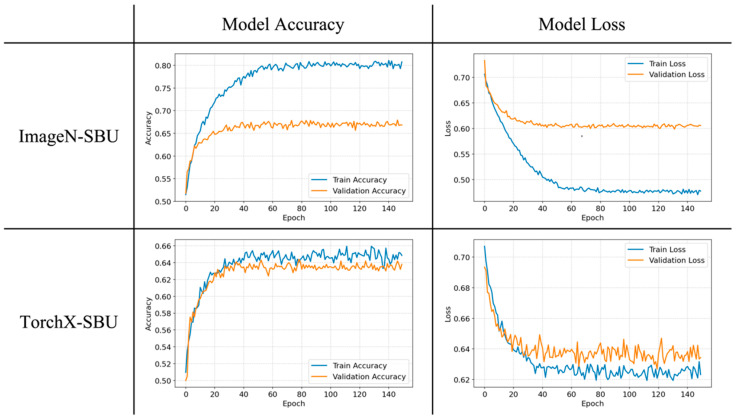
Learning curves of the best ImageN-SBU and TorchX-SBU model. For the ImageN-SBU model, the training accuracy stabilizes at a high level, while the validation accuracy plateaus at a lower level, suggesting potential overfitting. On the other hand, the TorchX-SBU model displays a closer alignment between training and validation accuracy, indicating a more generalized model performance.

**Figure 5 diagnostics-15-00845-f005:**
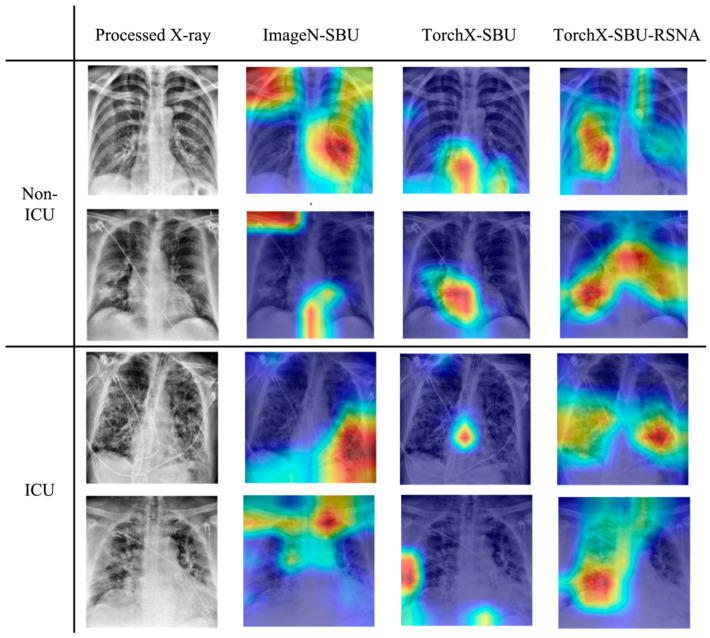
Grad-CAM heatmaps of the best ImageN-SBU, TorchX-SBU, and TorchX-SBU-RSNA models, demonstrating the focal areas significant for ICU admission classification. Each row represents processed X-ray images from non-ICU and ICU patients, respectively. The TorchX-SBU-RSNA model, enhanced with an extended dataset, shows a more targeted activation within the lung areas, minimizing attention to non-relevant regions like the neck, diaphragm, or shoulders, which are more pronounced in the other models. In the heatmaps, red regions indicate high importance for the model’s decision, while dark blue regions indicate low importance.

**Table 1 diagnostics-15-00845-t001:** Data splitting for ImageN-SBU and TorchX-SBU models.

Sets	Label	NY-SBU	Augmented
Train	ICU	172	403
	Non-ICU	403	403
Validation	ICU	43	101
	Non-ICU	101	101
Test	ICU	54	54
	Non-ICU	126	126

**Table 2 diagnostics-15-00845-t002:** Data splitting for TorchX-SBU-RSNA and ELIXR-SBU-RSNA models.

Sets	Label	NY-SBU	RICORD	Total	Augmented
Train	ICU	172	94	266	642
	Non-ICU	403	239	642	642
Validation	ICU	43	24	67	161
	Non-ICU	101	60	161	161
Test	ICU	54	-	54	54
	Non-ICU	126	-	126	126

**Table 3 diagnostics-15-00845-t003:** Comparative performance of different models.

Model	Precision	Recall	Specificity	Accuracy	Balanced Accuracy	AUC	PR AUC
ImageN-SBU	43.6%	55.6%	67.9%	64.2%	61.7%	0.657	0.426
TorchX-SBU	48.3%	58.9%	71.3%	67.6%	65.1%	0.711	0.488
TorchX-SBU-RSNA	55%	58%	80%	73.3%	68.9%	0.772	0.632
ELIXR-SBU-RSNA	55.8%	62.2%	77.5%	72.9%	69.8%	0.777	0.629

**Table 4 diagnostics-15-00845-t004:** Comparison of existing models for ICU admission prediction using chest X-ray images. * Globally Aware Multiple Instance Classifier (GMIC).

Study	Data	Model	Performance
Chamberlin et al. [30]	2456 chest X-rays (50% COVID-negative)	dCNNs + logistic regression (chest X-ray pre-trained)	AUC = 0.870
Li et al. [33]	8357 chest X-rays (all COVID patients)	DenseNet121 (chest X-ray pre-trained)	AUC = 0.76
Shamout et al. [32]	6449 chest X-rays (all COVID patients)	COVID-GMIC * (ChestX-ray14 pre-trained)	X-ray only: AUC = 0.738 PR AUC = 0.439 X-ray + laboratory tests: AUC = 0.786 PR AUC = 0.517
ImageN-SBU	COVID-19-NY-SBU (*n =* 899, all COVID patients)	DenseNet121 (ImageNet pre-trained)	AUC = 0.657 PR AUC = 0.426
TorchX-SBU	COVID-19-NY-SBU (*n =* 899, all COVID patients)	DenseNet121 (TorchXRayVision pre-trained)	AUC = 0.711 PR AUC = 0.488
TorchX-SBU-RSNA	COVID-19-NY-SBU and MIDRC-RICORD-1c (*n =* 1316, all COVID patients)	DenseNet121 (TorchXRayVision pre-trained)	AUC = 0.772 PR AUC = 0.632
ELIXR-SBU-RSNA	COVID-19-NY-SBU and MIDRC-RICORD-1c (*n =* 1316, all COVID patients)	ELIXR (chest X-ray pre-trained)	AUC = 0.777 PR AUC = 0.629

## Data Availability

The COVID-19-NY-SBU dataset can be accessed publicly via The Cancer Imaging Archive (TCIA) at https://www.cancerimagingarchive.net/collection/covid-19-ny-sbu/ (accessed on 27 June 2022). Additionally, the MIDRC-RICORD-1c dataset is also available via TCIA at https://www.cancerimagingarchive.net/collection/midrc-ricord-1c/ (accessed on 29 June 2022). The codes used to process the data and perform the analysis are available on GitHub at https://github.com/yunchibellalin/COVID-ICU.git.

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
