# Peer review of "Classification of the ICU Admission for COVID-19 Patients with Transfer Learning Models Using Chest X-Ray Images"

_diagnostics, 2025, doi:10.3390/diagnostics15070845_

Round 1
Reviewer 1 Report
Comments and Suggestions for Authors
Comments on the Manuscript Entitled “Prediction of the ICU Admission for COVID-19 Patients with Transfer Learning Models Using Chest X-ray Images”
- The Abstract lacks clarity in conveying the research motivation. Additionally, a more thorough explanation and introduction of the methodology are required.
- The Introduction does not adequately discuss the reviewed studies. The authors must address the limitations of the existing literature before highlighting the identified research gap. While Deep Learning is introduced, there is insufficient focus on Transfer Learning models, which is the central theme of the manuscript. The authors should explore the application of such models in the literature to better frame the research gap.
- The authors’ method for aligning the MIDRC-RICORD-1c dataset from multi-class to binary class labels, particularly concerning the pseudo-non-ICU group, requires further justification. The resulting pseudo binary labels appear to be clearly imbalanced.
- The motivation for selecting pre-trained models is not addressed in the manuscript.
- The use of a dropout rate of 0.3 in the trainable layers lacks justification and should be explained.
- In the Experiment section (2.5), the authors report different pre-trained models than those discussed in the previous section (2.3). This inconsistency needs clarification; for instance, ResNet50 is omitted here, and no model configurations are provided for this model.
- Figure 4 illustrates clear overfitting in the ImageN-SBU model. Although the authors acknowledge this modeling issue, it is not discussed or addressed adequately. This oversight renders the modeling process for this pre-trained model unstable, making the comparison and classification performance unfair. Further clarification and discussion are necessary.
- Including pseudo-code or an algorithm could enhance the clarity of the proposed methodology.
- There is a lack of robust comparison between the proposed method and existing state-of-the-art studies. Although the authors compare their results with some previous studies in the discussion section, a more comprehensive comparison in tabular form should be provided.
- The complexity of the methodology is not discussed. Given the data augmentation added and skewed data distribution, model complexity should be adequately addressed.
- The conclusion is underdeveloped, lacking depth and suggestions for future research directions.
Technical Comments
- I suggest changing “Prediction” to “Classification” throughout the manuscript.
- All abbreviations must be defined upon their first appearance.
- The referencing/citations style should be corrected, ensuring that reference numbers are included.
- The quality of Figure 1 should be improved for better clarity and presentation.
Author Response
Overview of the revision
We are grateful for the valuable comments from the three reviewers. We have carefully revised the manuscript based on the reviewer’s comments and address each comment with the responses listed below. One major criticism from the reviewers, is that the model performance has not been overly impressive. As pointed out by one reviewer, the precision was relatively low (54% in the first submission). In addition, compared to the previous two leading models, our model’s AUC (0.756) was slightly lower than Li et al.’s model (0.76) and Shamout et al.’s (0.786). To address this, the most major revision in the version is that we have refined our model design by switching backbone CNN to DenseNet121. Also, we have added another model developed by Google (ELIXR) which was pretrained with a large amount of X-ray. By making those changes, we have slightly improved the AUC from 0.756 to 0.777. The AUC demonstrated by our model is almost the same as the other two leading models which were both trained with much larger but private datasets than ours. Moreover, Shamout et al.’s best performing model was trained with X-ray images as well as the lab data. If only X-ray is available, their model would have a lower AUC (0.738) and PRAUC (0.439). Considering the limited data availability encountered in our study, we feel that the training strategies adopted in this work have provided a competitive and comparable performance to the current leading models. Such strategies may benefit future model training and development. The response to individual comments is listed below.
Comments 1: The Abstract lacks clarity in conveying the research motivation. Additionally, a more thorough explanation and introduction of the methodology are required.
Response 1: We have revised the abstract to clarify the research motivation and provide a more comprehensive introduction to the methodology.
Comments 2: The Introduction does not adequately discuss the reviewed studies. The authors must address the limitations of the existing literature before highlighting the identified research gap. While Deep Learning is introduced, there is insufficient focus on Transfer Learning models, which is the central theme of the manuscript. The authors should explore the application of such models in the literature to better frame the research gap.
Response 2: We appreciate the reviewer’s suggestions and have added a paragraph in the Introduction to highlight these gaps and further explain the relevance of Transfer Learning models.
Comments 3: The authors’ method for aligning the MIDRC-RICORD-1c dataset from multi-class to binary class labels, particularly concerning the pseudo-non-ICU group, requires further justification. The resulting pseudo binary labels appear to be clearly imbalanced.
Response 3: We appreciate the reviewer’s concern regarding the imbalance in our binary classification of RSNA airspace disease grading into ICU versus non-ICU labels. We have revised the manuscript in the Methods section (2.1.3). In brief, our approach is supported by evidence demonstrating that severe airspace disease (opacities in >4 lung zones) significantly increases the risk of ICU admission or mortality. Specifically, Hoang-Thi et al. (2022) found that patients with extensive lung involvement (e.g., highest quartile scores) have a markedly higher risk of ICU admission or death, while Au-Yong et al. (2022) reported similar outcomes for cases with >75% lung area affected. These findings provide evidence for classifying severe cases as pseudo-ICU and grouping milder cases (1–4 zones) as pseudo-non-ICU, reflecting clinical reality where severe cases, though less common, are critically distinct. The resulting imbalance mirrors real-world disease distribution, as supported by the literature. We have added this rationale to Section 2.1.3, citing the referenced studies, to clarify our methodology.
Comments 4: The motivation for selecting pre-trained models is not addressed in the manuscript.
Response 4: Previously, we focused on ResNet50 due to its strong performance in medical imaging AI. Based on the reviewer's suggestion, we also experimented DenseNet121, as used in leading studies (H. Li et al., 2023; Shamout, F.E., 2021) and did find it to perform slightly better. As a result, we have switched to DenseNet121 and updated our results accordingly in Table 3. Additionally, we implemented an advanced image encoder proposed by Google to explore whether language model-inspired techniques can enhance performance. The updated results are reported in Table 3.
Comments 5: The use of a dropout rate of 0.3 in the trainable layers lacks justification and should be explained.
Response 5: The choice of dropout rate was determined through hyperparameter fine-tuning, where we tested various combinations to identify the best-performing configuration. We have explained how this process was done in Section 2.5: Experiment.
Comments 6: In the Experiment section (2.5), the authors report different pre-trained models than those discussed in the previous section (2.3). This inconsistency needs clarification; for instance, ResNet50 is omitted here, and no model configurations are provided for this model.
Response 6: We thank the reviewer for pointing out this issue. We have revised the manuscript to make this more consistent. In Section 2.3, we introduced ResNet50 as the architecture of our pre-trained model, which we have now switched to DenseNet121 in this revision.
Comments 7: Figure 4 illustrates clear overfitting in the ImageN-SBU model. Although the authors acknowledge this modeling issue, it is not discussed or addressed adequately. This oversight renders the modeling process for this pre-trained model unstable, making the comparison and classification performance unfair. Further clarification and discussion are necessary.
Response 7: We thank the reviewer for pointing this out. We have tried to fine-tune the ImageN-SBU model and attempted to reduce its overfitting. We have added description for this effort by optimizing dropout, learning rate decay, batch normalization, and early stopping, as detailed in the 2.5 Experiment section. While these techniques reduced overfitting, they did not eliminate it entirely. We believe that this overfitting is likely due to the intrinsic differences in data characteristics between natural images and X-rays. Overfitting is a known challenge when applying ImageNet-pretrained models to specific medical images. Studies have shown that features important in natural images (textures, colors, object boundaries) differ significantly from those in medical imaging, which rely more on shapes, patterns, and grayscale variations (Yang Wen et al. and Tauhidul Islam et al.). As a result, ImageNet models may overemphasize irrelevant features, leading to poor generalization. Additionally, X-ray datasets—especially those for specific conditions like COVID-19—tend to be smaller and less diverse compared to ImageNet. This difference could worsen overfitting, as a model trained on an extensive and varied dataset like ImageNet may not be able to adapt to a more homogeneous, smaller medical dataset. This is why we compared ImageNet pretraining to an X-ray-pretrained model. As shown in our results, the X-ray-pretrained model has reduced the overfitting, leading to more stable performance. This underscores the importance of medical-specific pretraining in deep learning to address the limitations of cross-domain transfer learning. We have also added a paragraph addressing the overfitting issue in the Discussion section.
Comments 8: Including pseudo-code or an algorithm could enhance the clarity of the proposed methodology.
Response 8: We appreciate the reviewer’s suggestion. However, the codes are rather complicated and it could be difficult to present the methodology as pseudo-codes or algorithm in a clear and succinct way. We will share our processing and training codes on GitHub for interested researchers to review and/or test with their data. Currently, the repository is set as private as we are still in the revision process, but we will make it public as soon as we are close to finalizing the work.
Comments 9: There is a lack of robust comparison between the proposed method and existing state-of-the-art studies. Although the authors compare their results with some previous studies in the discussion section, a more comprehensive comparison in tabular form should be provided.
Response 9: We thank the reviewer for pointing this out and have strengthened the discussion. We have also compiled a comparison of currently leading models and our models in Table 4 for a more comprehensive evaluation.
Comments 10: The complexity of the methodology is not discussed. Given the data augmentation added and skewed data distribution, model complexity should be adequately addressed.
Response 10: We have added details on computational and optimization complexity in Experiment section 2.5 to better address the model complexity.
Comments 11: The conclusion is underdeveloped, lacking depth and suggestions for future research directions.
Response 11: We are grateful for the suggestion and have revised the Conclusion. Suggestions for future research directions have been added too.
Technical comments 1: I suggest changing “Prediction” to “Classification” throughout the manuscript.
Technical response 1: We are now using ‘Classification’ throughout the manuscript, as the reviewer suggested.
Technical comments 2: All abbreviations must be defined upon their first appearance.
Technical response 2: We apologize for the errors and have checked all abbreviations to make sure they are defined upon their first appearance in the manuscript.
Technical comments 3: The referencing/citations style should be corrected, ensuring that reference numbers are included.
Technical response 3: We apologize for missing this issue previously and have corrected the referencing style.
Technical comments 4: The quality of Figure 1 should be improved for better clarity and presentation.
Technical response 4: We have updated all figures, including Figure 1, with higher-resolution versions for improved clarity and presentation.
References
Hoang-Thi, T. N., D. T. Tran, H. D. Tran, M. C. Tran, T. M. Ton-Nu, H. M. Trinh-Le, H. N. Le-Huu, N. M. Le-Thi, C. T. Tran, N. N. Le-Dong, and A. T. Dinh-Xuan. "Usefulness of Hospital Admission Chest X-Ray Score for Predicting Mortality and Icu Admission in Covid-19 Patients." J Clin Med 11, no. 12 (2022).
Au-Yong, I., Y. Higashi, E. Giannotti, A. Fogarty, J. R. Morling, M. Grainge, A. Race, I. Juurlink, M. Simmonds, S. Briggs, S. Cruickshank, S. Hammond-Pears, J. West, C. J. Crooks, and T. Card. "Chest Radiograph Scoring Alone or Combined with Other Risk Scores for Predicting Outcomes in Covid-19." Radiology 302, no. 2 (2022): E11.
Wen, Y., Chen, L., Deng, Y., & Zhou, C. (2021). Rethinking pre-training on medical imaging. Journal of Visual Communication and Image Representation, 78, 103145.
Islam, T., Hafiz, M. S., Jim, J. R., Kabir, M. M., & Mridha, M. F. (2024). A systematic review of deep learning data augmentation in medical imaging: Recent advances and future research directions. Healthcare Analytics, 100340.
Shamout, F. E., Y. Shen, N. Wu, A. Kaku, J. Park, T. Makino, S. Jastrzebski, J. Witowski, D. Wang, B. Zhang, S. Dogra, M. Cao, N. Razavian, D. Kudlowitz, L. Azour, W. Moore, Y. W. Lui, Y. Aphinyanaphongs, C. Fernandez-Granda, and K. J. Geras. "An Artificial Intelligence System for Predicting the Deterioration of Covid-19 Patients in the Emergency Department." NPJ Digit Med 4, no. 1 (2021): 80.
Li, H., K. Drukker, Q. Hu, H. M. Whitney, J. D. Fuhrman, and M. L. Giger. "Predicting Intensive Care Need for Covid-19 Patients Using Deep Learning on Chest Radiography." J Med Imaging (Bellingham) 10, no. 4 (2023): 044504.
Reviewer 2 Report
Comments and Suggestions for Authors
The paper addresses the prediction ability of transfer learning model for ICU admission of COVID-19 patients.The author should focus on the following points for improvement of the manuscript.
1.Author should mention the objective of the paper in the introduction section point wise.
2.The performance parameter values obtained from confusion matrices are very low.For example AUC obtained for image N-SBU is 0.673,for TorchX-SBU-RSNA it is 0.756.Author should improve the accuracy and AUC value of each model and they should try for implimentation of other DL models such as ViT, VGG-19 for improving the performance .
3.Author should improve the resolution of Fig.4.
Comments on the Quality of English LanguageEnglish language can be improved.
Author Response
Overview of the revision
We are grateful for the valuable comments from the three reviewers. We have carefully revised the manuscript based on the reviewer’s comments and address each comment with the responses listed below. One major criticism from the reviewers, is that the model performance has not been overly impressive. As pointed out by one reviewer, the precision was relatively low (54% in the first submission). In addition, compared to the previous two leading models, our model’s AUC (0.756) was slightly lower than Li et al.’s model (0.76) and Shamout et al.’s (0.786). To address this, the most major revision in the version is that we have refined our model design by switching backbone CNN to DenseNet121. Also, we have added another model developed by Google (ELIXR) which was pretrained with a large amount of X-ray. By making those changes, we have slightly improved the AUC from 0.756 to 0.777. The AUC demonstrated by our model is almost the same as the other two leading models which were both trained with much larger but private datasets than ours. Moreover, Shamout et al.’s best performing model was trained with X-ray images as well as the lab data. If only X-ray is available, their model would have a lower AUC (0.738) and PRAUC (0.439). Considering the limited data availability encountered in our study, we feel that the training strategies adopted in this work have provided a competitive and comparable performance to the current leading models. Such strategies may benefit future model training and development. The response to individual comments is listed below.
Comments 1: Author should mention the objective of the paper in the introduction section point wise.
Response 1: We appreciate the reviewer’s suggestion and have revised the Introduction to explicitly state the objectives of the paper in a point-wise format. These can be found in the final paragraph of the Introduction section.
Comments 2: The performance parameter values obtained from confusion matrices are very low. For example AUC obtained for imageN-SBU is 0.673,for TorchX-SBU-RSNA it is 0.756. Author should improve the accuracy and AUC value of each model and they should try for implimentation of other DL models such as ViT, VGG-19 for improving the performance.
Response 2: We are grateful for the reviewer’s comment. Although we could not find a publicly available chest X-ray pretrained ViT/VGG-19 model as suggested, in this revision, we have implemented DenseNet121, which is used in leading studies we referenced (Shamout et al. and Li et al.). DenseNet121 has provided a slightly improvement to the performance (AUC from 0.756 to 0.772), which is comparable to the leading models in literature (AUC 0.738 from Shamout et al. and AUC 0.76 from Li et al.). Consequently, we have switched our model architecture to DenseNet121 and updated our results to the manuscript accordingly.
Beyond conventional CNNs, we also explored a recent advanced model, ELIXR, proposed by Google, which leverages language-model-inspired embedding techniques (Andrew B. Sellergren et al., Shawn Xu et al.). This model achieved an AUC of 0.78 in our prediction task. We have revised the Discussion and added more interpretation about the performance of our model. We understand that an AUC of 0.78 may not seem clinically satisfactory, so we have provided explanations that our main contribution through this work is to show that training strategies with X-ray pretrained models and data expansion techniques are useful for future model development and refinement.
Comments 3: Author should improve the resolution of Fig.4.
Response 3: We have updated Figure 4 with higher-resolution images.
References
Shamout, F. E., Y. Shen, N. Wu, A. Kaku, J. Park, T. Makino, S. Jastrzebski, J. Witowski, D. Wang, B. Zhang, S. Dogra, M. Cao, N. Razavian, D. Kudlowitz, L. Azour, W. Moore, Y. W. Lui, Y. Aphinyanaphongs, C. Fernandez-Granda, and K. J. Geras. "An Artificial Intelligence System for Predicting the Deterioration of Covid-19 Patients in the Emergency Department." NPJ Digit Med 4, no. 1 (2021): 80.
Li, H., K. Drukker, Q. Hu, H. M. Whitney, J. D. Fuhrman, and M. L. Giger. "Predicting Intensive Care Need for Covid-19 Patients Using Deep Learning on Chest Radiography." J Med Imaging (Bellingham) 10, no. 4 (2023): 044504.
Sellergren, A.B., Chen, C., Nabulsi, Z., Li, Y., Maschinot, A., Sarna, A., Huang, J., Lau, C., Kalidindi, S.R., Etemadi, M., Garcia-Vicente, F., Melnick, D., Liu, Y., Eswaran, K., Tse, D., Beladia, N., Krishnan, D., Shetty, S., 2022. Simplified Transfer Learning for Chest Radiography Models Using Less Data. Radiology 305, 454–465.. https://doi.org/10.1148/radiol.212482
Shawn Xu et al. ELIXR: Towards a general purpose X-ray artificial intelligence system through alignment of large language models and radiology vision encoders arXiv:2308.01317v2
https://doi.org/10.48550/arXiv.2308.01317
Reviewer 3 Report
Comments and Suggestions for Authors
1. Inconsistence use of abbreviations, like FDA and ICU never abbreviated before if first used in the body part.
2. How could more samples (~7000) be a research limitation even if the deep learning model is better for more data?
3. The author explains the higher number of private data is a research limitation, but the author augmented it to balance the data, which increases the number of data. Which directly contradicts the author's claim.
4. There is no explanation, of how the author classifies poor and high-quality samples.
5. No flow direction in Figure 1 claims and is difficult to understand at the first gland.
6. Figure 4 shows a significant gap between training and validation accuracy for ImageN-SBU (natural image pre-training), indicating overfitting. The authors do not address this critical limitation.
7. The TorchX-SBU-RSNA model achieves an AUC of 0.756 and precision of 0.544, both of which are subpar compared to existing studies (e.g., Shamout et al.: AUC=0.786 with clinical data; Li et al.: AUC=0.76–0.78). A precision of 54.4% implies that nearly half of the predicted ICU cases are false positives, rendering the model clinically unreliable for resource allocation.
8. The authors neither justify this poor performance nor address how such high false-positive rates could strain ICU resources during outbreaks. This lack of critical discussion undermines the model’s practical utility.
9. Weak comparison to literature. The discussion cites Chamberlin et al. (AUC=0.87) but dismisses their work as "two-step" without explaining why a less accurate "one-step" model is preferable.
10. Inconsistent abbreviations terms like "ICU," "pseudo-ICU," and "non-ICU" are used interchangeably without clear definitions, causing confusion in interpretation.
Oversampling via rotation (±15°) is simplistic and insufficient for addressing inherent variability in severe vs. non-severe cases
Author Response
Overview of the revision
We are grateful for the valuable comments from the three reviewers. We have carefully revised the manuscript based on the reviewer’s comments and address each comment with the responses listed below. One major criticism from the reviewers, is that the model performance has not been overly impressive. As pointed out by one reviewer, the precision was relatively low (54% in the first submission). In addition, compared to the previous two leading models, our model’s AUC (0.756) was slightly lower than Li et al.’s model (0.76) and Shamout et al.’s (0.786). To address this, the most major revision in the version is that we have refined our model design by switching backbone CNN to DenseNet121. Also, we have added another model developed by Google (ELIXR) which was pretrained with a large amount of X-ray. By making those changes, we have slightly improved the AUC from 0.756 to 0.777. The AUC demonstrated by our model is almost the same as the other two leading models which were both trained with much larger but private datasets than ours. Moreover, Shamout et al.’s best performing model was trained with X-ray images as well as the lab data. If only X-ray is available, their model would have a lower AUC (0.738) and PRAUC (0.439). Considering the limited data availability encountered in our study, we feel that the training strategies adopted in this work have provided a competitive and comparable performance to the current leading models. Such strategies may benefit future model training and development. The response to individual comments is listed below.
Comments 1: Inconsistence use of abbreviations, like FDA and ICU never abbreviated before if first used in the body part.
Response 1: We apologize for missing those in the first version and have checked that all abbreviations are defined upon their first appearance in the manuscript.
Comments 2: How could more samples (~7000) be a research limitation even if the deep learning model is better for more data?
Response 2: We apologize for the confusion and revised the manuscript so that we do not describe the large datasets as limitation. What we meant to say is that, for the currently leading models, large and private datasets were used in their training. For researchers who do not have access to such private datasets, the lack of access is a limitation for developing and validating AI models. We have revised the manuscript to clarify this point of view.
Comments 3: The author explains the higher number of private data is a research limitation, but the author augmented it to balance the data, which increases the number of data. Which directly contradicts the author's claim.
Response 3: Again we apologize for this confusion. As described in the previous comment, we now do not describe the large datasets as limitation themselves. The limitation was meant to describe the lack of data availability for researchers that do not have access to the private datasets. As for the data augmentation, it is adopted as an approach to expand the dataset when the access to private large datasets is limited, and it could be regarded as a strategy to overcome the limitations of data availability issues. We have revised the manuscript to clarify on this.
Comments 4: There is no explanation, of how the author classifies poor and high-quality samples.
Response 4: We have revised the Datasets section to explain the judgement. In this study, image quality screening was done manually and we have excluded images that: 1. Were captured in a lateral view instead of the AP (anterior-to-posterior) or PA (posterior-to-anterior) view 2. Were cropped or incomplete, meaning they did not contain the full lung fields necessary for assessment. This has also been clarified in the updated exclusion flowchart.
Comments 5: No flow direction in Figure 1 claims and is difficult to understand at the first gland.
Response 5: We have updated Figure 1 with clear flow direction.
Comments 6: Figure 4 shows a significant gap between training and validation accuracy for ImageN-SBU (natural image pre-training), indicating overfitting. The authors do not address this critical limitation.
Response 6: We thank the reviewer for pointing this out and have revised the manuscript accordingly. We thank the reviewer for pointing this out. We have tried to fine-tune the ImageN-SBU model and attempted to reduce its overfitting. We have added description for this effort by optimizing dropout, learning rate decay, batch normalization, and early stopping, as detailed in the 2.5 Experiment section. While these techniques reduced overfitting, they did not eliminate it entirely. We believe that this overfitting is likely due to the intrinsic differences in data characteristics between natural images and X-rays. Overfitting is a known challenge when applying ImageNet-pretrained models to specific medical images. Studies have shown that features important in natural images (textures, colors, object boundaries) differ significantly from those in medical imaging, which rely more on shapes, patterns, and grayscale variations (Yang Wen et al. and Tauhidul Islam et al.). As a result, ImageNet models may overemphasize irrelevant features, leading to poor generalization. Additionally, X-ray datasets—especially those for specific conditions like COVID-19—tend to be smaller and less diverse compared to ImageNet. This difference could worsen overfitting, as a model trained on an extensive and varied dataset like ImageNet may not be able to adapt to a more homogeneous, smaller medical dataset. This is why we compared ImageNet pretraining to an X-ray-pretrained model. As shown in our results, the X-ray-pretrained model has reduced the overfitting, leading to more stable performance. This underscores the importance of medical-specific pretraining in deep learning to address the limitations of cross-domain transfer learning. We have also added a paragraph addressing the overfitting issue in the Discussion section.
Comments 7: The TorchX-SBU-RSNA model achieves an AUC of 0.756 and precision of 0.544, both of which are subpar compared to existing studies (e.g., Shamout et al.: AUC=0.786 with clinical data; Li et al.: AUC=0.76–0.78). A precision of 54.4% implies that nearly half of the predicted ICU cases are false positives, rendering the model clinically unreliable for resource allocation.
Comments 8: The authors neither justify this poor performance nor address how such high false-positive rates could strain ICU resources during outbreaks. This lack of critical discussion undermines the model’s practical utility.
Response 7 / 8 : We appreciate the reviewer’s comments and have updated the manuscript accordingly. To improve our model performance, in this revision, we have switched our model architecture to DenseNet121, achieving an AUC of 0.77. We have also additionally implemented the Google ELIXR model, which utilizes an image encoder to generate embeddings for classification, achieving an AUC of 0.78. Although the improvement is not superb, we are now providing similar performances to the leading models of Shamout et al. and Li et al. under a smaller training dataset. We have also added PR AUC to evaluate model performance on the trade-off between recall and precision and to better compare with prior studies that also reported this metric.
Regarding low precision, we noticed that even the leading model (Shamout et al.) experience this issue. They reported the best precision as 23.8% with a PR AUC of 0.517. In comparison, our model achieved a higher precision of 55% with a PR AUC of 0.632. While the precision could be affected by data imbalance, we acknowledge the concern regarding clinical reliability. To enhance clinical utility, future work integrating diverse data (e.g., lab results, demographics) might reduce false positives. This discussion has been added to the revised manuscript.
Comments 9: Weak comparison to literature. The discussion cites Chamberlin et al. (AUC=0.87) but dismisses their work as "two-step" without explaining why a less accurate "one-step" model is preferable.
Response 9: We appreciate the reviewer’s comment and have revised the discussion accordingly. We have removed the text saying that a ‘two-step’ approach is a limitation or less desirable. We are now pointing out that this model included data of non-COVID patients (about 50%), so the prediction performance is not directly comparable to our, Li’s or Shamout’s models that exclusively focus on COVID-19 patients.
Comments 10: Inconsistent abbreviations terms like "ICU," "pseudo-ICU," and "non-ICU" are used interchangeably without clear definitions, causing confusion in interpretation.
Response 10: We appreciate the reviewer’s comment and have replaced the ‘pseudo-ICU’ notation with ‘ICU’.
Comments 11: Oversampling via rotation (±15°) is simplistic and insufficient for addressing inherent variability in severe vs. non-severe cases.
Response 11: We appreciate the reviewer’s comment. The main reason we only chose to perform data augmentation with rotation is two fold. First, some other data augmentation strategies, such as shifting or contrast adjustment, would have their effects canceled by our image processing such as the lung segmentation and histogram equalization process. Second, for some strategies such as adding random noise, we are not sure what a feasible level of data augmentation shall be performed. We are also concerned that adding random noise may potentially mislead the model given that our prediction task is sensitive to subtle lung changes. For instance, overly aggressive noise might resemble opacities or consolidation, altering the original pathological meaning of the sample. As a result, we only attempted a simple augmentation with rotation. We have discussed potential advanced data augmentation techniques in the manuscript for reference (last paragraph of the Discussion section).
Reference
Wen, Yang, Leiting Chen, Yu Deng, and Chuan Zhou. "Rethinking Pre-Training on Medical Imaging." Journal of Visual Communication and Image Representation 78 (2021): 103145.
Islam, Tauhidul, Md Sadman Hafiz, Jamin Rahman Jim, Md Mohsin Kabir, and M. F. Mridha. . "A Systematic Review of Deep Learning Data Augmentation in Medical Imaging: Recent Advances and Future Research Directions." Healthcare Analytics 100340. (2024).
Shamout, F. E., Y. Shen, N. Wu, A. Kaku, J. Park, T. Makino, S. Jastrzebski, J. Witowski, D. Wang, B. Zhang, S. Dogra, M. Cao, N. Razavian, D. Kudlowitz, L. Azour, W. Moore, Y. W. Lui, Y. Aphinyanaphongs, C. Fernandez-Granda, and K. J. Geras. "An Artificial Intelligence System for Predicting the Deterioration of Covid-19 Patients in the Emergency Department." NPJ Digit Med 4, no. 1 (2021): 80.
Li, H., K. Drukker, Q. Hu, H. M. Whitney, J. D. Fuhrman, and M. L. Giger. "Predicting Intensive Care Need for Covid-19 Patients Using Deep Learning on Chest Radiography." J Med Imaging (Bellingham) 10, no. 4 (2023): 044504.
Round 2
Reviewer 3 Report
Comments and Suggestions for Authors
N/A